# Correlation Between Red Cell Distribution Width and Peripheral Vascular Disease Severity and Complexity

**DOI:** 10.3390/medsci7070077

**Published:** 2019-07-09

**Authors:** Seçkin Satılmış, Ahmet Karabulut

**Affiliations:** 1Department of Cardiology, Acibadem Atakent Hospital, Istanbul 34303, Turkey; 2Department of Cardiology, Acibadem Mehmet Ali Aydınlar University School of Medicine, Istanbul 34752, Turkey

**Keywords:** red cell distribution width, atherosclerosis, peripheral vascular disease, biomarker

## Abstract

A traditional hematological marker, red cell distribution width (RDW), is accepted as a novel marker of atherosclerotic vascular diseases. Clinical importance of the RDW as a prognostic biomarker in peripheral vascular disease (PVD) has been reported in a few studies. Herein, we aimed to show the correlation between RDW and PVD severity and its complexity in terms of angiographic evaluation. A total of 118 patients who underwent peripheral lower extremity angiography were subsequently evaluated retrospectively. Upon admission, RDW level was measured with automated complete blood count. Severity and complexity of the PVD was evaluated according to Trans-Atlantic Inter-Society Consensus (TASC II) classification. A TASC II A-B lesion was defined as simple PVD, and a TASC II C-D lesion was defined as prevalent and complex PVD. Then, both groups were compared statistically according to clinical, laboratory, and demographic features, including RDW levels. In 49.6% of the patients, TASC II C-D lesions were observed. Advanced age, male gender, and body mass index (BMI) were associated with TASC II groups. Red cell distribution width levels were correlated with presence of PVD, as well as TASC II grades (p:0.02). The fourth quartile (75th percentile) of the RDW levels was 14.1, and patients with RDW levels ≥14.1 had a more significant correlation with the presence and severity of PVD (p:0.001). In the multivariate regression analysis, elevated RDW was found to be an independent predictor of the presence of PVD and also TASC II C-D lesions (OR:2.26, with a 95% confidence interval (CI) 0.051–0.774; p:0.02). Elevated RDW levels was associated with TASC II C-D lesions, which indicated more prevalent and complex PVD.

## 1. Background

Red cell distribution width (RDW) is defined as the quotient of standard deviation of erythrocyte volume and its mean volume, which is expressed as a percentage. It is a measure of variability in size of the circulating erythrocytes, and it is accepted as a traditional marker in hematological diseases, particularly anemia and bone marrow dysfunction [1]. It is easily assessable in routine automated complete blood count analysis. Clinical usefulness of the RDW has increased in the last 10 years, since it was reported that RDW could be a prognostic factor in the various diseases, as well as all-cause mortality [1]. Correlation between RDW and atherosclerotic vascular disease was clearly reported in numerous studies [2,3,4,5,6,7,8,9,10,11,12,13,14]. The indicator of atherosclerotic process, carotid plaque and increased carotid intima/media thickness, in asymptomatic patients was associated with higher RDW percentile [2]. The RDW level was also associated with carotid atherosclerotic progression, and higher RDW levels were linked to future risk of incident stroke [3,4,5]. The correlation between RDW and coronary atherosclerosis was also clearly demonstrated [1,6]. Subjects with higher RDW levels tend to have higher mortality regarding coronary artery disease [7,8,9]. There was also linkage between the extent and complexity of the coronary artery disease and RDW levels. Higher RDW levels were correlated with more complex and severe coronary artery disease [10,11,12,13,14]. 

Peripheral vascular disease (PVD) of the lower limb is a spectrum atherosclerotic disease, ranging from mild plaque formation to chronic total vessel occlusion. Patients may remain asymptomatic in mild forms of the disease but can develop intermittent claudication, rest pain, or tissue loss, including ulceration and gangrene, as the disease progresses. Clinical presentation is usually correlated with the extent of vascular involvement. Correlation of RDW and PVD was investigated in a few reports [15,16,17]. Higher RDW levels were associated with increased all-cause mortality and more prevalent PVD, which was assessed with ankle-brachial index [16]. The predictive value of RDW on the severity and complexity of PVD was not reported clearly. Herein, we aimed to demonstrate the association between RDW and the severity of PVD, which was determined by the angiography of lower extremity.

## 2. Methods

### 2.1. Patient Selection

A total of 118 patients who had underwent peripheral lower extremity angiography were subsequently included in this study. Patients were evaluated retrospectively and clinical risk factors, such as medical history, laboratory results, and peripheral angiography recordings, was entered into computerized database. Exclusion criteria was as follows: Acute coronary syndromes, patients with baseline anemia, history of blood transfusion within the last three months, active infection, chronic inflammatory diseases, malignancy, and decompensated heart failure.

### 2.2. Procedure and Protocol

Upon admission, patients were evaluated with anamnesis, and physical examination and blood samples were taken for analysis. Peripheral angiography was performed with a 6-French pigtail catheter using an automatic pump injector. For the evaluation of both lower extremities, the catheter tip was positioned above the aorto-iliac bifurcation. The Trans-Atlantic Inter-Society Consensus (TASC) II score analysis (Table 1) was performed on bilateral aorto-iliac arterial segments [18]. The patients’ angiographic data were evaluated from catheter laboratory records by two interventional cardiologists and the TASC II grade was noted for each patient. Red cell distribution width analysis was performed using the automated hematology analyzer Sysmex XT-1800i (Roche Diagnostic, Istanbul, Turkey) with an 11.5–15.3% normal range. 

### 2.3. Definitions

Severity and complexity of the PVD was evaluated according to angiographic TASC II classification (Table 1). Anemia was defined as Hemoglobin (Hb) < 13 g/dL in men and 12 g/dL in women, according to World Health Organization criteria. Diabetes mellitus (DM) was defined according to the patients’ history, use of insulin or anti-diabetic agents, fasting glucose >126 mg/dL, or random blood glucose level >200 mg/dL. Dyslipidemia was defined as either low density lipoprotein (LDL) cholesterol >100 mg/dL or triglycerides >150 mg/dL, or both, or drug use for dyslipidemia. Hypertension (HT) was defined as previous use of anti-hypertensive medications or a systolic pressure >140 mmHg or a diastolic pressure >90 mmHg on at least two separate measurements. Smoking was defined as current regular use of cigarettes or cigars within the last six months.

## 3. Statistical Analysis and Approval of the Study

Statistical analyses were performed using SPSS 21.0 (SPSS Inc., Chicago, IL, U.S.) software. The data are presented as mean ± SD with 95% confidence intervals (CI). The Student *t*-test was used for continuous variables between groups. Categorical variables were compared using the chi-square test and Fisher’s exact test. Correlation between RDW and PVD was demonstrated with Pearson’s correlation analysis. In addition, univariate and multivariate logistic regression analysis was performed to detect independent factors affecting severity of PVD. All *p* values were two-sided in the tests and *p* values less than 0.05 were considered to be statistically significant. 

The study was approved by the Local Ethics Committee of Acibadem University Atakent Hospital (02/01/2018). 

## 4. Results

The baseline demographic and clinical characteristics of the patients are summarized in Table 2. The mean age was 58.55 ± 11.07, ranging from 31 to 83 years, and 93 patients (79.5%) were male. The prevalence of HT and DM were 30.4% and 34.2%, respectively. The smoking status was relatively low despite the male predominance (35.3%). The mean RDW was 13.64 ± 1.33% and quartiles of the RDW levels were 12.8%, 13.5%, and 14.1%. Elevated RDW levels were more common in the older aged male gender. Red cell distribution width levels were prominently lower in the normal group and higher in the PVD (+) group (12.9 ± 0.79 vs 13.9 ± 1.39, *p* < 0.001). There was also a significant correlation between RDW and presence of PVD, as well as TASC II stages (Table 2). TASC II C-D lesions were associated with upper quartiles of RDW levels (Table 2). Multivariate logistic regression analysis, including all the demographic and clinical variables to detect predictors of severity and complexity of PVD, was performed (Table 3). Elevated RDW levels were the independent predictor for the PVD and TASC II C-D lesions. (OR:2.26, with a 95% CI 0.051–0.774; *p*:0.02).

## 5. Discussion

In the present study, we showed that RDW might be a prognostic factor in the atherosclerotic PVDs. Higher RDW levels were obtained in the TASC II C-D lesions which indicate the more prevalent and complex vascular involvement. These findings indicate that RDW is a significant clinical prognostic factor in the atherosclerotic vascular process, including coronary and carotid, as well as peripheral vascular beds.

Atherosclerosis is a leading cause of vascular disease worldwide. Its major clinical manifestations include ischemic heart disease, ischemic stroke, and peripheral arterial disease. Ischemic heart disease and ischemic stroke are the leading cause of mortality in the world [19]. Peripheral vascular disease is one of the major causes of disability. It is also a risk factor for cardiovascular mortality, and death linked to PVD was increased in the last 20 years. The traditional risk factor for atherosclerosis is age, hypertension, diabetes, dyslipidemia, obesity, and smoking [20]. The effective management of this traditional factors does not exclude the atherosclerotic progression. Chronic inflammation was accepted as a one of the major mechanisms in the progression of atherosclerosis. So, inflammatory markers, such as C-reactive protein (CRP), interlekukin (IL), and hematologic markers, were searched as an etiology or prognostic factor in the atherosclerosis, or both [21,22]. Atherosclerosis is often generalized. However, clinical manifestation of atherosclerotic involvement in different vascular systems may variate. Because of these variation, prognostic effect of inflammatory markers was checked in separate ways with numerous studies. 

Prognostic value of hematological indices was evaluated in the different atherosclerotic settings [1,2,4,6,15,23]. Although their sensitivity and specificity is moderate, availability of the test in the almost all clinics, fast and reliable results, and cost affordability make these indices popular prognostic clinical parameters. Hematological indices, predominantly RDW, neutrophil-lymphocyte ratio, and mean platelet volume, reflect the oxidative stress and inflammatory state which are postulated for the atherosclerotic process [6]. Previous studies showed that there was a correlation between RDW and inflammatory markers, such as interleukin-6, soluble tumor necrosis factor 1 and 2, fibrinogen, sedimentation rate, and high sensitivity CRP [6,9]. On the other hand, there is a close relationship between oxidative stress and RDW. Oxidative stress is responsible for shortening the lifespan of erythrocytes, which leads to both production and release of young cellular forms into the circulation [6]. Besides those mechanisms, RDW could also play a direct role in the microvascular dysfunction. Decreased RDW levels were associated with reduced whole blood viscosity. So, diminution of RBC deformability may impair the flow through the microcirculation [6,17]. Since 2007, numerous studies have been performed regarding the prognostic value of RDW. Red cell distribution width was found to be a predictor of cardiovascular and all-cause mortality [8]. Patients with a higher RDW percentile tend to have more prevalent atherosclerotic process. Red cell distribution width was a predictor for acute coronary syndromes and also chronic stable coronary heart disease. Red cell distribution predicts both short term and long-term mortality in vascular diseases [1,6,7,8,16,23].

The correlation between RDW and PVD was evaluated with a few investigations [15,16,17,24]. Ye et al. followed the patients with PVD defined by ankle-brachial index (ABI) for five and a half years. The mortality rate was highest in the patients with the highest quartile of RDW (>14.5) [16]. Zalawadiya et al. evaluated 6950 participants with ankle-brachial index. They found that higher RDW quartiles was associated with lower ABI and more severe PVD [15]. Demirtas et al. evaluated 82 patients with PVD in terms of Fountaine classification. The Fountain stage 3 group revealed higher RDW levels, which indicates the significant correlation of PVD severity and RDW [24]. All these studies preferred non-invasive tests to assess PVD severity. However, ABI, Fountaine, and Rutherford classification may not show the extent of the atherosclerotic process clearly when compared to angiographic assessment. Angiography could be gold standard to evaluate severity of PVD. A TASC II classification is an internationally-derived, collaboratively-created consensus definition that is used for the assessment of PVD according to the anatomic distribution and number and nature of lesions (stenosis, occlusion). A TASC II classification reflects the complexity and severity of the PVD more accurately. TASC II C and TASC II D groups indicate a more complex and more diffused atherosclerotic involvement. Interventional treatment of the TASC II C-D lesions are also challenged when compared to TASC II A-B [18]. In our study, we evaluated the patients as a simple (TASC II A-B) and complex (TASC II C-D) group in order to obtain more accurate statistical results. The analysis confirmed that complex lesions were associated with upper RDW quartiles. Further studies with a larger patient sample would show the correlation of RDW with each of the TASC II groups more accurately.

## 6. Limitations

Retrospective design and small sample size are the major limitations of the study. The inflammatory mediators, postulated as a major mechanism of RDW and atherosclerosis, including C-reactive protein, brain natriuretic peptide, erythropoietin, nitric oxide, fibrinogen, and cytokines, were not measured. The markers of the oxidative stress were also not evaluated. The other clinical classification representing the severity of PVD was not performed. The comments were done according to angiographic TASC II classification.

## 7. Conclusions

Red cell distribution is a cheap and widely reachable prognostic biomarker in atherosclerotic vascular disease. Similar to atherosclerotic coronary disease, elevated RDW levels was associated with TASC II C-D lesions, which indicated a more prevalent and complex PVD.

## Figures and Tables

**Table 1 medsci-07-00077-t001:** TASC II classification for the assessment of severity and complexity of peripheral vascular diseases (PVDs).

	AORTOILIAC LESIONS	FEMOROPOPLITEAL LESIONS
TASC II A	Single stenosis (<3 cm in length) in the CIA or EIA (unilateral/bilateral)	Single stenosis (<3 cm in length) in the superficial femoral artery or popliteal artery
TASC II B	1. Single stenosis (3–10 cm in length) not extending into the CFA 2. Heavily calcified stenosis up to 3 cm in length 3. Unilateral CIA occlusion	1. Single stenosis (3–10 cm in length) not involving distal popliteal artery 2. Heavily calcified stenosis up to 3 cm in length 3. Multiple lesions, each <3 cm in length (stenoses or occlusions) 4. Single or multiple lesions in the absence of continuous tibial runoff to improve inflow for distal surgical bypass
TASC II C	1. Bilateral stenosis (5–10 cm in length) in the CIA and/or EIA, not extending into the CFA 2. Multiple stenoses or occlusions (each 3–5 cm in length) 2. Unilateral EIA occlusion not extending into the CFA with or without heavy calcification 3. Unilateral EIA stenosis extending into the CFA 4. Bilateral CIA occlusion	1. Single stenosis or occlusion >5 cm in length 2. Unilateral EIA occlusion not extending into the CFA with or without heavy calcification
TASC II D	1. Diffuse, multiple unilateral stenosis involving the CIA, EIA, and CFA (usually >10 cm in length) 2. Unilateral occlusion involving both the CIA and EIA 3. Bilateral EIA occlusions 4. Diffuse disease involving the aorta and both iliac arteries 5. Iliac stenosis in a patient with abdominal aortic aneurysm or other lesions requiring aortic or iliac surgery	Complete CFA or superficial femoral artery occlusion or complete popliteal and proximal trifurcation occlusions

TASC II—Trans-Atlantic Inter-Society Consensus-2; CIA—common iliac artery; CFA—common femoral artery; EIA—external iliac artery.

**Table 2 medsci-07-00077-t002:** Distribution of clinical and demographic characteristics of the patients according to presence and severity of PVD.

Variables	PVD (−)(*n*:34)	PVD (+) (*n*:84)		*p*
		TASC II A-B (*n*:26) TASC II C-D (*n*:58)		
Age (years)	53.5 ± 10.8	58.5 ± 12.4	61.5 ± 9.5	0.003
Sex				0.001
Male	57.6% (19)	80.8% (21)	91.4% (53)	
Female	22.9% (14)	11.6% (5)	21.6% (5)	
Presence of Diabetes mellitus	31.3% (10)	28.0% (7)	31.0% (18)	0.95
Presence of Hypertension	48.5% (16)	26.9% (7)	29.3% (17)	0.12
Presence of Dyslipidemia	34.4% (11)	23.1% (6)	24.1% (14)	0.51
Current smoking	43.8% (14)	38.5% (10)	29.3% (17)	0.36
Creatinine (mg/dL)	0.89 ± 0.18	0.86 ± 0.30	1.15 ± 0.95	0.11
BMI (kg/m^2^)	25.9 ± 2.4	28.1 ± 4.7	28.5 ± 4.3	0.009
RDW (%)	12.9 ± 0.8	13.7 ± 1.1	14.0 ± 1.4	0.001
4th RDW quartile (≥14.1%)	2.9% (1)	34.6% (9)	34.5% (20)	0.002
Total	28.2%	22.2%	49.6%	

BMI—Body mass index. *P*< 0.05 is indicated as significant.

**Table 3 medsci-07-00077-t003:** Multivariate regression analysis for the reveal the predictor of PVD severity and complexity.

Dependent Variable: TASC II group	Odds Ratio	95% Confident Interval	*p*
Age (years)	2.62	0.005; 0.035	0.01
Sex (Male/Female)	0.63	−0.096; 0.186	0.52
Presence of Diabetes mellitus	1.27	−0.124; 0.567	0.20
Presence of Hypertension	−2.63	−0.862; − 0.120	0.01
Presence of Dyslipidemia	−0.68	−0.513; 0.250	0.49
Current smoking	0.07	−0.346; 0.374	0.93
Creatinine (mg/dL)	0.84	−0.130; 0.321	0.40
BMI (kg/m^2^)	2.46	0.009; 0.086	0.01
4th RDW quartile (>14.1%)	2.26	0.051; 0.774	0.02

*P* < 0.05 is indicated as significant.

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
