# Peer review of "Correlation Between Red Cell Distribution Width and Peripheral Vascular Disease Severity and Complexity"

_medsci, 2019, doi:10.3390/medsci7070077_

Reviewer 1 Report

The retrospective nature of the study may have affected the interpretation of the result but overall it needs few areas of clarification:

1)Why was other markers not simultaneously assessed?

2)Why was ABI not included

3)I am not sure of the statement on page 6  line 3: "Angiography could be gold standard to evaluate severity of PVD" is a correct statement as there are many variables including clinical, imaging and hemodynamic criteria.

4)Grammar and spelling corrections required 

Author Response

Dear Reviewer,

Thanks for your valuable comments.

- As you expressed, retrospective nature of the article is the major limitation of the the study. Because of this, only avaliable data was included to study. The recordings about ABI and othe laboratory markers were missing.

- Angiographic evaluation of the peripheral vascular disease may give exact anatomical involvement of the disease. Hovewer, it does not mean the severity of clinical presentation. In our study, we analysed the anatomical involvement of the lower extremity vessels accordin to TASC classification.

- Grammar and spelling corrections done by proof-reading company from London.

Reviewer 2 Report

The authors have demonstrated a correlation between red cell distribution width and peripheral vascular disease. The study is well presented and I appreciate the authors for addressing an important topic. Please see below my comments and suggestions to improve the manuscript.

Minor comments:

1.       I understand that patients with anemia were excluded from the study. However, there could have been patients who were in the pathological progress towards anemia. Could the authors discuss (or include in the results) the data on patients getting anemic after being recruited into the study and how this could have affected the correlation outcome?

2.       Most of the introduction revolves around RDW. Could the authors include background information regarding severity of PVD and why it needs to be addressed?

3.       I recommend using a language editor such as grammarly to improve English errors throughout the manuscript

4.       Please include the name of the ethics committee that approved the study.

Major comments:

1.       Since RDW may increase during iron deficiency and chronic liver diseases, do the authors have these data in the selected patients? If so, I would suggest doing a multivariate analysis and justify the claim that RDW was increased due to PVD rather than the underlying deficiencies.

2.       Could the authors include data on MCV (if available) as it may provide further clarification and reliability of the current data?

3.       Odds ratio of hypertension is negative. Does this mean HT is protective of PVD? I believe this could be due to consolidation of different grades of hypertension. I would suggest subcategorizing hypertension into different grades before doing a multivariate analysis.

4.       Since RDW has been recently shown to be associated with certain forms of cancer, could the authors include cancer data of included patients and perform a multivariate analysis?

Author Response

Dear Reviewer,

Thanks for your valuable comments.

-All the patients with anemia, liver disease and cancer was excluded from the study. The MCV levels did not evaluated within the statistical analysis. Because, MCV levels were within the normal range (80-90).

-In this article, our aim was to evaluate predictive value of the RDW in the severity of peripheral vascular disease. Thus, we primarily focused on the predictive value of the RDV in the different settings of atherosclerotic vascular diseases. General information about the severity of peripheral vascular disease was indicated in the Background section.

-The name of the Ethic Committes was indicated within the article.

-Grammar and spelling corrections done by proof-reading company from London.